# Active Learning for Affinity Prediction of Antibodies

**Alexandra Gessner**[1]* **Sebastian W. Ober**[2]* **Owen Vickery**[2]* **Dino Oglić**[1] **Talip Uçar**[1]

[1]Centre for AI, BioPharma R&D, [2]Biologics Engineering, Oncology R&D, AstraZeneca
{first_name.last_name}@astrazeneca.com

## Abstract

The primary objective of most lead optimization campaigns is to enhance the binding affinity of ligands. For large molecules such as antibodies, identifying mutations that enhance antibody affinity is particularly challenging due to the combinatorial explosion of potential mutations. When the structure of the antibody-antigen complex is available, relative binding free energy (RBFE) methods can offer valuable insights into how different mutations will impact the potency and selectivity of a drug candidate, thereby reducing the reliance on costly and time-consuming wet-lab experiments. However, accurately simulating the physics of large molecules is computationally intensive. We present an active learning framework that iteratively proposes promising sequences for simulators to evaluate, thereby accelerating the search for improved binders. We explore different modeling approaches to identify the most effective surrogate model for this task, and evaluate our framework both using pre-computed pools of data and in a realistic full-loop setting.

## 1 Introduction

Proteins are complex macromolecules composed of linear chains of amino acids folded into a tertiary structure. Antibodies are noteworthy proteins that interact with a diverse range of molecules via their complementarity-determining regions (CDRs). Through a process called *sonic hypermutation*, the immune system continually refines and optimizes the specificity and strength of antibody binding to a particular antigen via iteration of mutation and selection. We can emulate such a biological process by applying active learning to relative binding free energy (RBFE) methods, which model the antibody-antigen interaction to predict their binding affinity relative to a parent antibody.

Active learning is a model-based strategy to taking informed decisions about future experiments to perform under a certain objective. A special case is Bayesian optimization (Garnett, 2023) that uses this approach for global optimization, making it a suitable method for identifying improved binders using RBFE methods.

The primary goal of this work is the systematic identification of novel antibody mutations that lead to improved binding properties when forming a complex with a particular antigen, with the following contributions. For related work, please refer to Appendix A.1.

1. We construct a Bayesian optimization loop that interacts with RBFE methods to identify antibody sequences with enhanced stability and improved binding properties w.r.t. a wild type antibody.

2. We introduce a simple yet effective encoding scheme based on the BLOcks SUbstitution Matrix (BLOSUM) (Henikoff & Henikoff, 1992b,a) and evaluate it together with other encoding schemes.

3. We validate our method on pre-calculated data and evaluate a range of antibody sequence encodings and model choices.

---

*equal contribution

Workshop on Bayesian Decision-making and Uncertainty, 38th Conference on Neural Information Processing Systems (NeurIPS 2024).

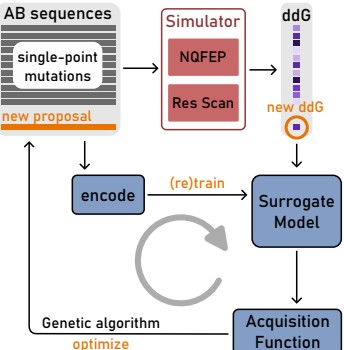

Figure 1: Schematic of the active learning loop. We initially train the surrogate model with a set of single-point mutations of our wild-type antibody for which we have results from an RBFE simulator (Section 2.1). We run Bayesian optimization to sequentially propose new sequences that improve binding affinity. The new candidate is proposed by optimizing an acquisition function and triggers a new query of the simulator. The encoded sequence and the simulated output are added to the pool and used to update the surrogate model. The building blocks of the active learning loop are discussed in Section 2.2

4. We run the most promising setups from the validation runs in a full loop and find that our algorithm consistently finds vastly improved binders over the mutations available in the pre-computed dataset.

# 2 Methods

## 2.1 Relative binding free energy methods

Relative binding free energy (RBFE) simulators are powerful methods for the prediction of binding affinity, incorporating protein flexibility, solvents, and entropic effects. Despite their limitations due to high computational costs, force field accuracy, and automation challenges, RBFE methods offer valuable insights into protein-protein interactions, binding affinity, and specificity.

We employ two different RBFE simulators with varying degrees of dynamics to estimate the relative binding affinity $\Delta\Delta G$ of an antibody that has been mutated with respect to a fixed parent and target, based on their sequences of amino acids. The RBFE simulators compute $\Delta\Delta G = (\Delta G_{\mathrm{bound}}) - (\Delta G_{\mathrm{unbound}})$, where $(\Delta G_{\mathrm{bound}})$ and $(\Delta G_{\mathrm{unbound}})$ represent the difference in Gibbs free energy between the wild type (WT) and the mutant in the bound and unbound states respectively. $\Delta\Delta G$ hence denotes the gain or loss of energy from forming an antibody-antigen complex relative to the WT, and a negative $\Delta\Delta G$ indicates an enhancement in binding affinity.

**NQFEP:** Utilizing our own implementation of the non-equilibrium free energy perturbation (NQFEP) method with GROMACS and pmx (Seeliger & De Groot, 2010), we calculate the $\Delta G$ of a specific mutation in both the bound and unbound states. By completing two legs of the thermodynamic cycle, we calculate the $\Delta\Delta G$ of the mutation. This simulator provides fairly accurate estimates but at a high computational cost of 6-24 hours on a modern GPU.

**Schrödinger Res Scan:** This cheaper (∼minutes on a few CPUs) but less accurate alternative estimates the absolute energies of the bound complex and the unbound states, taking the $\Delta\Delta G$ as the difference between these absolute energies. This method is error-prone, particularly because it uses a single snapshot of the complex.

## 2.2 Active learning loop

**Bayesian optimization** Bayesian optimization (BO; Garnett, 2023) is a black-box optimization technique aimed at finding the minimum (w.l.o.g.) of a function $f : \mathcal{X} \to \mathbb{R}$, where in our case $\mathcal{X}$ represents the space of antibodies, and $f$ denotes the true $\Delta\Delta G$. The method uses observed input-output pairs $(\mathbf{x}_i, y_i)$ to construct a probabilistic *surrogate model*. The model uncertainty enables an efficient exploration-exploitation trade-off through an acquisition function $a : \mathcal{X} \to \mathbb{R}$ that aims to estimate the utility of selecting a new antibody and its simulated $\Delta\Delta G$ value. Given an initial set of sequences and $\Delta\Delta G$ values, we alternate between updating the surrogate model and acquiring new sequences until our budget (e.g., time, or total compute) is exhausted.

The most common class of surrogate models for BO are Gaussian processes (GPs; Rasmussen & Williams, 2006). We model the observations as $y_i = f(\mathbf{x}_i) + \epsilon_i$, with $\epsilon_i \sim \mathcal{N}(0, \sigma^2)$, and place a GP prior on $f$. A GP is defined entirely by its mean function, $\mu : \mathcal{X} \to \mathbb{R}$ (here a learned constant), and its kernel function, $k : \mathcal{X} \times \mathcal{X} \to \mathbb{R}$, which measures the similarity between two points.

**Encoding antibody sequences**    GPs do not naturally operate on sequence data. In order to use them on antibody sequences, we need to map strings of amino acids $s \in \mathcal{S}$ to numerical values $\mathbf{x} \in \mathcal{X}$, $\text{emb}(s) : s \mapsto \mathbf{x}$. We compare the following embeddings:

**One-hot encoding:** Each letter in the alphabet of amino acids gets assigned a unique one-hot vector and the encoding is their concatenation according to the sequence.

**Bag of amino acids:** We count matching $n$-grams of amino acids, corresponding to the bag of words embedding (Jurafsky & Martin, 2000). We set $n = 5$.

**BLOSUM:** The Blocks Substitution Matrix (BLOSUM; Henikoff & Henikoff, 1992a) is a substitution matrix used in bioinformatics for sequence alignments of proteins. It is an indefinite similarity matrix quantifying the similarity between pairs of amino acids upon substitution. We perform an eigendecomposition of this kernel matrix, $U D U^\top$, and encode the individual amino acids with the rows of $U|D|^{1/2}$. This is motivated by observations in Oglic & Gärtner (2018, 2019) on flip-spectrum transformations of indefinite kernels.

**AbLang2:** AbLang2 (Olsen et al., 2024) uses an antibody-specific language model to encode the light and heavy chain of the antibody jointly.

**Gaussian processes on sequence embeddings**    We consider two types of GP models that are apt to handle the high-dimensional sequence embeddings; firstly, dot-product covariance functions, where we focus on the Tanimoto kernel (Ralaivola et al., 2005),

$$k_{\text{Tanimoto}}(\mathbf{x}, \mathbf{x}') = \frac{\langle \mathbf{x}, \mathbf{x}' \rangle}{\|\mathbf{x}\| + \|\mathbf{x}'\| - \langle \mathbf{x}, \mathbf{x}' \rangle}, \tag{1}$$

and secondly, stationary kernels. For the latter, we use the RBF and Matérn-$3/2$ kernels after projecting the embedded data to a lower-dimensional space where these covariance functions are known to work better. For the dimension reduction we use random matrices of size $N_{\text{emb}} \times N_{\text{low}}$ with entries sampled from a normal distribution, partly inspired by Wang et al. (2016).

**Active learning on sequence data**    In order to propose new mutations of the parental antibody for querying the simulator, we require a mechanism for optimizing the acquisition function. To do so, we follow the work of Moss et al. (2020), and use a genetic algorithm (GA), which "evolves" a "population" of sequences to maximize the "fitness" of the sequences, in this case the acquisition value. The evolution is achieved by a *mutation* operation, which introduces a single amino acid mutation, and a *crossover* operation, which takes two existing sequences, splices them at a random location, and combines the splices to create a new sequence. In practice, to implement our genetic algorithm, we extend the `mol_ga` package (Tripp & Hernández-Lobato, 2023) (based off the packages of Brown et al., 2019; Jensen, 2019) to handle mutation and crossover operations for protein sequences.

## 3    Data

We pre-simulated $\Delta\Delta G$ values for selected mutations of a parent antibody, including $532$ single-point mutations from the costly NQFEP simulator and $60,479$ mutations from the Schrödinger Res Scan simulator (cf. Section 2.1). The input data comprises sequences of amino acids for both the light and heavy chains of the antibody. We use the standard single-letter codes representing the 20 amino acids. Except for AbLang2, all embeddings concatenate the light and heavy chains using an auxiliary character to treat them as a single sequence. The resulting sequences have a fixed length of 238, obviating the need for padding. The embeddings are of dimensions of 4998 for the one-hot and BLOSUM encodings, 835 (NQFEP) and 834 (Schrödinger Res Scan) for the bag-of-amino-acids employing 5-grams, respectively, and 480 for AbLang2.

## 4    Experiments

### 4.1    Validation

To validate our method, we run the BO loop on both pre-simulated datasets for different embedding schemes and choices of GP model. We initialize the GP with a small subset of the data, $0.01\%$

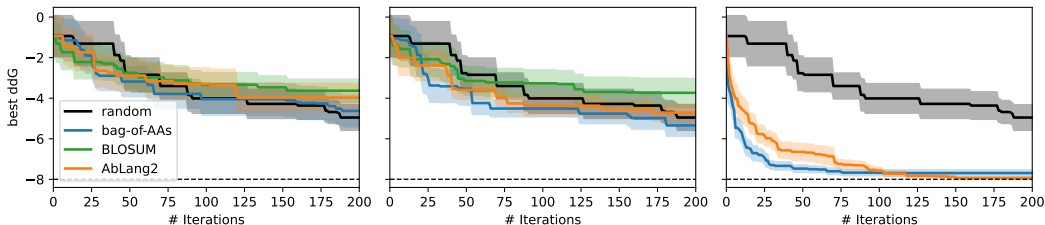

Figure 2: Validation on the Schrödinger Res Scan pre-computed dataset over 200 iterations averaged over 10 runs. Best $\Delta\Delta G$ value found using the RBF *(left)*, Matérn *(center)*, and Tanimoto *(right)* kernels, respectively, for all encodings. In the case of the RBF and Matérn kernel, the embeddings have been projected to 5 dimensions. The horizontal dashed line is the best value in the dataset.

$(0.001\%)$ of the NQFEP (the Schrödinger Res Scan) data. In each iteration of the Bayesian optimization loop, we evaluate the acquisition function on the heldout data and find the new candidate by choosing the maximizing value. We use expected improvement from `BoTorch` (Balandat et al., 2020) as acquisition function and fix the observation variance $\sigma^2$ to $10^{-4}$. Fixing the noise variance is motivated by the determinism of the Schrödinger Res Scan simulations and the low variance expected for the NQFEP simulations. The experiments for NQFEP can be found in Appendix A.2. For selected choices of covariance function and embedding, we run 10 trials with 200 loop iterations each and track the best value found by the algorithm. Our baseline is a random strategy that records the best $\Delta\Delta G$ observed from random picks of a (sequence, $\Delta\Delta G$)-pair from the remaining pool.

Due to the size of the Schrödinger Res Scan dataset, we restrict the selection of model settings and compare the Tanimoto and stationary kernels for all embeddings except the one-hot encoding, and fix the dimension reduction to five. The results for fixed noise variance are displayed in Figure 2; those for trained noise variance can be found in Appendix A.2.2. While the stationary kernels do not even outperform the random strategy, the Tanimoto kernel with the AbLang2 encoding finds the best value in the dataset ($-8.00 \, \mathrm{kcal/mol}$) within $\approx 150$ iterations. For the Tanimoto kernel, the BLOSUM encoding did not terminate in validation mode due to the large embedding size.

## 4.2 Full loop

We use the results from Section 4.1 to inform model choices when placing the simulator in the loop. Due to computational constraints, we only consider the Schrödinger Res Scan simulator. We select the AbLang2, BLOSUM, and bag-of-AAs encodings and choose the Tanimoto kernel for the GP and initialize the loop using single-residue mutants of the wild type, choosing 3 random mutations at each residue in the antibody's CDR. We then run the BO loop for 50 iterations, using the GA-based acquisition to select new queries from the set of all possible CDR mutations to send to the simulator. Each experiment is repeated 3 times. The results are shown in Section 4.2. We observe that for all encodings, using an unrestricted search space allows us to quickly find significantly better $\Delta\Delta G$ values than using an explicit pool of allowed sequences. Interestingly, the AbLang2 encoding now shows the weakest performance of all the methods, suggesting that it might be spending more resources exploring the space than the other encodings, as the space is extremely large.

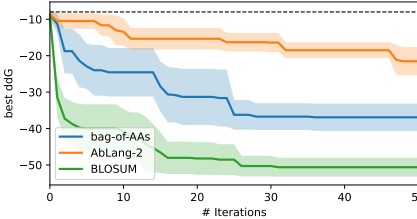

Figure 3: Results of the full loop run with the Schrödinger Res Scan simulator. We plot the best $\Delta\Delta G$ values found for each of three encodings using the Tanimoto kernel, as well as the best value from the pooled data from the validation experiments as a dashed line.

# 5 Conclusion and Outlook

Our work employs active learning to accelerate the in silico search for antibody mutations with improved binding properties to a fixed antigen. We studied various choices of sequence encodings and model choices for Bayesian optimization in sequence space and found well-performing model choices. In the future, we aim to include structural information about the antibody and adapt the model to handle multiple data sources. Finally, we plan to validate it on experimental data.

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

# A Additional information

## A.1 Related work

**Bayesian optimization** has been used to optimize sequences in both discrete sequence space (Lodhi et al., 2002; Moss, 2023) and continuous latent space (Stanton et al., 2022; Gómez-Bombarelli et al., 2018). For example, substring kernels combined with a black-box optimization method such as GA are proposed for optimization in discrete space (Moss, 2023; Khan et al., 2022) while the optimization in latent space is done through embeddings of sequences in a generative modelling setting (Grosnit et al., 2021; Deshwal & Doppa, 2021).

**Scoring functions** used for optimizing molecules for binding affinity include docking scores (Noh et al., 2022; Jeon & Kim, 2020), relative binding free energy (RBFE) (Ghanakota et al., 2020; Moore et al., 2023) and absolute binding free energy (ABFE) (Feng et al., 2022; Eckmann et al., 2024, 2022), and can be used as a proxy for binding affinity. Moreover, the docking score is known to be inaccurate and has low correlation with experimentally measured affinities (Coley et al., 2020; Handa et al., 2023; Pinzi & Rastelli, 2019) while scores based on binding free energy tend to be more accurate, but computationally more expensive, than docking scores (Moore et al., 2023). Since we focus on lead optimization in this work, we use RBFE simulators as they tend to be computationally more efficient.

## A.2 Validation experiments

The validation loop, described in Section 4.1 is conceptually visualized in Figure 4.

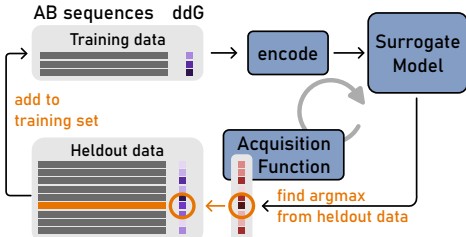

Figure 4: Schematic of the validation loop. We split the pre-computed dataset into a training set and a pool of held-out data. The loop iteratively selects a new sequence from the pool by maximizing the acquisition function on the held-out data and updates the surrogate model.

### A.2.1 Validation experiments for NQFEP

For the smaller dataset from the NQFEP simulator, we ran the validation for all combinations of GP models with all encodings. For the latter, we additionally explore the effect of the dimension we project the embedded sequences to prior to feeding them into the kernel. We record the best $\Delta\Delta G$ value found by the Bayesian optimization routine up to the current iteration.
Overall, we observe that the choice of projection dimension has little influence on the performance of the BO model. The choice of encoding appears to be more important where AbLang2 consistently outperforms the other encodings. This is especially true for the Tanimoto kernel (right panel of Figure 5), where the best value in the dataset ($-5.3\,\mathrm{kcal/mol}$) is found within 30 iterations across all ten trials.

**Fixed noise variance** Figure 5 shows the results for the RBF, Matérn, and Tanimoto kernel, where the input to the stationary dimensions has been projected to five dimensions. We observe that the Tanimoto kernel clearly outperforms the stationary kernels. Figure 6 and Figure 7 show additional results for the RBF and Matérn kernel, respectively, for different projection dimensions for the embeddings. Only the AbLang2 encoding consistently outperforms the random strategy. At larger projection dimensions, the one-hot encoding also displays good performance.

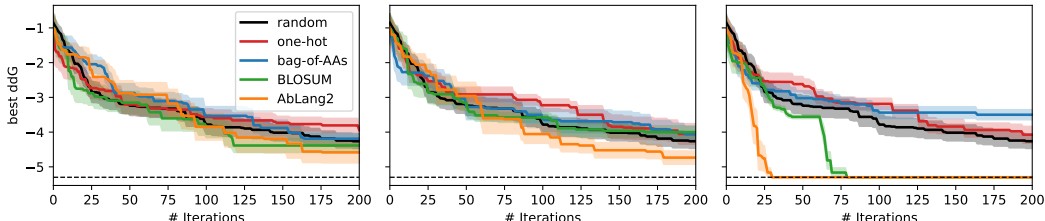

Figure 5: Validation on the NQFEP pre-computed dataset over 200 iterations averaged over 10 runs. Best $\Delta\Delta G$ value found using the RBF *(left)*, Matérn *(center)*, and Tanimoto *(right)* kernels, respectively, for all encodings. In the case of the RBF and Matérn kernel, the embeddings have been projected to 5 dimensions. The horizontal dashed line is the best value in the dataset.

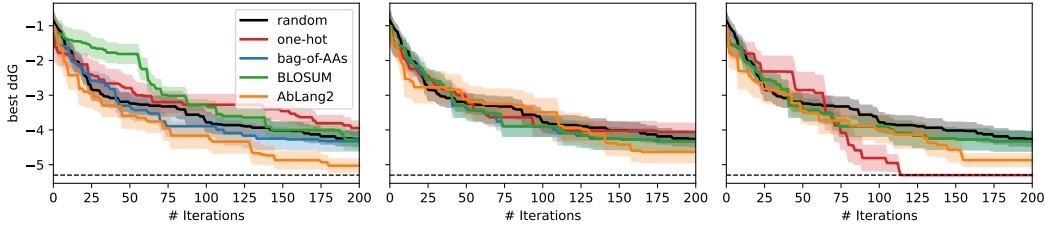

Figure 6: Validation on the NQFEP pre-computed dataset over 200 iterations averaged over 10 runs. Best $\Delta\Delta G$ value found using the RBF kernel with inputs projected to 10 *(left)*, 15 *(center)*, and 20 *(right)* dimensions using random projections, respectively, for all encodings. The horizontal dashed line is the best value in the dataset.

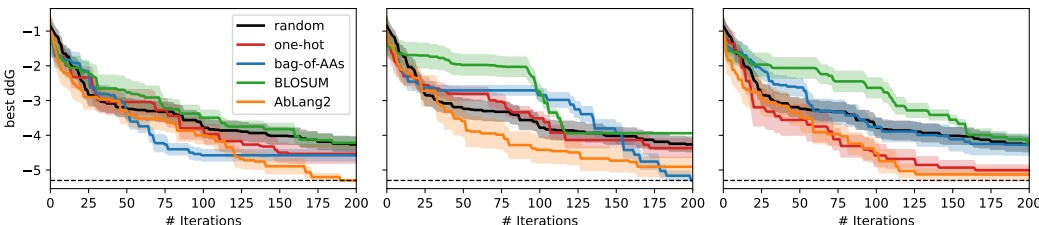

Figure 7: Validation on the NQFEP pre-computed dataset over 200 iterations averaged over 10 runs. Best $\Delta\Delta G$ value found using the Matérn kernel with inputs projected to 10 *(left)*, 15 *(center)*, and 20 *(right)* dimensions using random projections, respectively, for all encodings. The horizontal dashed line is the best value in the dataset.

**Trained noise variance** We present the same results as before but train the variance of the likelihood. This requires a modification of the acquisition function in the BO loop, since expected improvement assumes noise-free evaluations. Instead, we employ noisy expected improvement using quasi Monte Carlo (Letham et al., 2018), and rely on the implementation in `BoTorch`. The results in Figure 8, Figure 9, and Figure 10 are organized as their counterparts for fixed noise variance in Figure 5, Figure 6, and Figure 7.

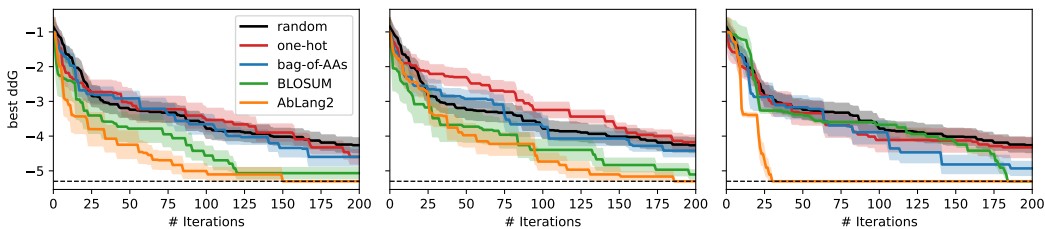

Figure 8: Same as Figure 5, but with learned noise variance, RBF *(left)*, Matérn *(center)*, and Tanimoto *(right)* kernel.

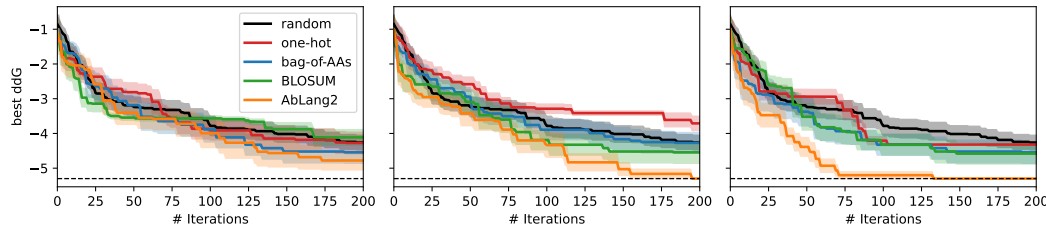

Figure 9: Same as Figure 6, but with learned noise variance, for the RBF kernel

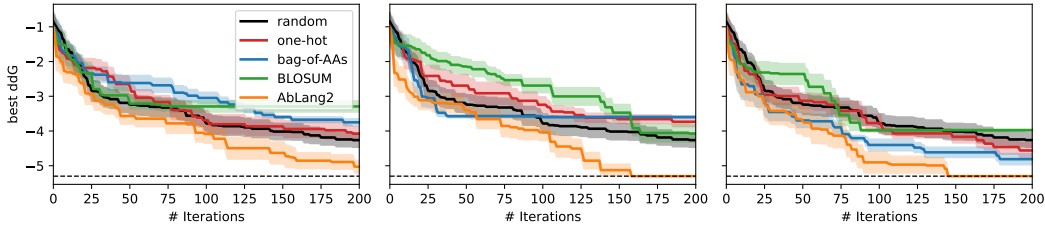

Figure 10: Same as Figure 7, but with learned noise variance, for the Matérn kernel.

### A.2.2 Schrödinger Res Scan with learned noise variance

Figure 11 shows the results for the larger dataset when training the variance of the likelihood. Remarkably, the RBF kernel combined with AbLang2 (left panel of Figure 11) performs significantly better than in the case of fixed noise. However, the training becomes unstable and the runs did not actually complete. This is indicated by the plateau of the curve. For the Tanimoto kernel, only the AbLang2 encoding completed its runs. Due to the larger size of the other encodings, the evaluation of the acquisition function on the entire dataset becomes prohibitively expensive.

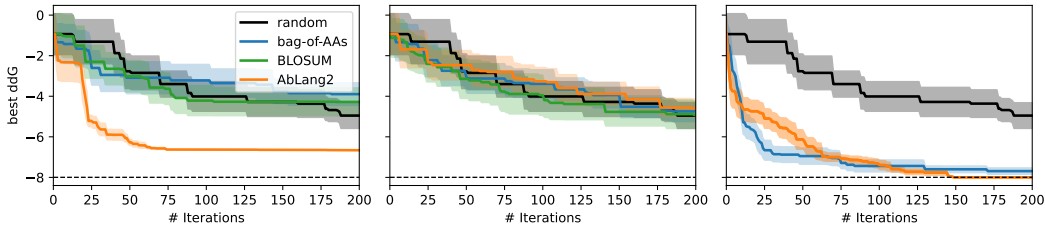

Figure 11: Validation on the Schrödinger Res Scan pre-computed dataset. Same as Figure 2, but with learned noise variance, for the RBF *(left)*, Matérn *(center)*, and Tanimoto *(right)* kernel.

