# OpenReview forum: "Active Learning for Affinity Prediction of Antibodies"
_NeurIPS.cc/2024/Workshop/BDU — NeurIPS BDU Workshop 2024 Poster_

### Official Review · Reviewer_H4Gr · 2024-09-21
**The paper proposes active learning for affinity prediction of antibodies, but there are minor shortcomings.**

**Rating:** 7
**Confidence:** 3

**Review:**

This paper presents a novel active learning framework that iteratively proposes promising sequences for simulators to evaluate, and conduct experiments with different modeling approaches to enhance the search for improved binders.

#### **Strength**

1) The affinity prediction of antibodies is an interesting field and the exepriments are solid of pre-computed pools of data and a realistic full-loop setting.
2) This paper proposes an active learning framework which will reduce the reliance on costly and time-consuming wetlab experiments.

#### **Weakness**

1) The authors do not provide some key parameters for the Gaussian Process (GP) kernel on Page 3, line 90. For example, the value of gamma in the RBF kernel is missing.

2) The paper mentions that the AbLang2 encoding method tends to consume excessive resources in exploring the search space (Page 4, line 143). Similarly, on Page 4, line 131, it is stated that the BLOSUM encoding did not terminate in validation mode due to the large embedding size when using the Tanimoto kernel. Given the availability of numerous dimensionality reduction techniques (UMAP, Autoencoder, Random Projection), it would be beneficial to explore such methods to mitigate these issues and improve computational efficiency.

---

### Official Review · Reviewer_Dcw7 · 2024-10-04
**Review of "Active Learning for Affinity Prediction of Antibodies"**

**Rating:** 7
**Confidence:** 3

**Review:**

**Summary:** This work introduces a novel application of active learning in the context of antibody affinity optimization by integrating Bayesian optimization with the RBFE method for optimizing binding affinity. The combination of these techniques can improve the efficiency of evaluating antibody mutations and reduce computational costs.

**Pros:** In addition to the novelty mentioned in my summary, the visual representation in Figure 1 is particularly helpful for understanding the active learning framework.

**Cons:**
- As someone more familiar with Bayesian optimization than biology, I found it challenging to translate biological terms in this paper into the language of Bayesian optimization. For example, the numbers 532, 60,479, 20, 238, and 4,998 in the experimental section were not clearly explained in the context of Bayesian optimization, making it difficult to fully understand the experiment setup. Additionally, the dimension of the search space was unclear to me based on the description in the experimental section.
- This paper lacks a detailed explanation of the traditional RBFE methods. Are they simply random sampling? While the paper mentions the improved efficiency of the proposed framework as its advantage, it would benefit from a direct comparison between the process of traditional methods and the active learning (Bayesian optimization) framework.
- The choice of the genetic algorithm for optimizing the acquisition function is not clearly justified in the paper. While a genetic algorithm is a reasonable choice for discrete sequence spaces, the advantages of this method over alternatives (e.g., random search, gradient-based methods) are not discussed.
- Given the high dimensionality in the context of this paper, I believe there would be a significant performance improvement by replacing the expected improvement acquisition function with alternative acquisition functions or techniques that are better suited to higher dimensions. The paper could consider discussing such alternatives.

---

### Decision · Program_Chairs · 2024-10-09

Accept (Poster)